# Effects and Causes of Detraining in Athletes Due to COVID-19: A Review

**DOI:** 10.3390/ijerph19095400

**Published:** 2022-04-28

**Authors:** Alfredo Córdova-Martínez, Alberto Caballero-García, Enrique Roche, Daniel Pérez-Valdecantos, David C. Noriega

**Affiliations:** 1Department of Biochemistry, Molecular Biology and Physiology, Faculty of Health Sciences, GIR Physical Exercise and Aging, University of Valladolid, Campus Duques de Soria, 42004 Soria, Spain; danielperezvaldecantos@gmail.com; 2Department of Anatomy and Radiology, Faculty of Health Sciences, GIR Physical Exercise and Aging, University of Valladolid, Campus Los Pajaritos, 42004 Soria, Spain; alberto.caballero@uva.es; 3Department of Applied Biology-Nutrition, Institute of Bioengineering, University Miguel Hernández, 03202 Elche, Spain; eroche@umh.es; 4Institute for Health and Biomedical Research (ISABIAL), 03010 Alicante, Spain; 5CIBER Fisiopatología de la Obesidad y Nutrición (CIBEROBN), Instituto de Salud Carlos III (ISCIII), 28029 Madrid, Spain; 6Department Cirugía, Oftalmología, Otorrinolaringología y Fisioterapia, Facultad de Medicina, 47003 Valladolid, Spain; dcnoriega1970@gmail.com; 7Departamento de Columna Vertebral, Hospital Clínico Universitario de Valladolid, 47003 Valladolid, Spain

**Keywords:** athletes, clinical consequences, COVID-19, physical activity, return to sport, SARS-CoV-2

## Abstract

Several aspects of systemic alterations caused by the SARS-CoV-2 virus and the resultant COVID-19 disease have been currently explored in the general population. However, very little is known about these particular aspects in sportsmen and sportswomen. We believe that the most important element to take into account is the neuromuscular aspect, due to the implications that this system entails in motion execution and coordination. In this context, deficient neuromuscular control when performing dynamic actions can be an important risk factor for injury. Therefore, data in this review refer mainly to problems derived in the short term from athletes who have suffered this pathology, taking into account that COVID-19 is a very new disease and the presented data are still not conclusive. The review addresses two key aspects: performance alteration and the return to regular professional physical activity. COVID-19 causes metabolic-respiratory, muscular, cardiac, and neurological alterations that are accompanied by a situation of stress. All of these have a clear influence on performance but at the same time in the strategy of returning to optimal conditions to train and compete again after infection. From the clinical evidence, the resumption of physical training and sports activity should be carried out progressively, both in terms of time and intensity.

## 1. Introduction

The COVID-19 pandemic has affected many people in general and athletes in particular. This has led to a series of restrictions, which from a pathophysiological point of view, may affect the athlete’s performance in the short and long term. The restrictions basically affect training and eating habits, disturbing physical condition, as well as psychological behavior and general health status [1,2,3,4,5].

The disease caused by SARS-CoV-2 infection, known as COVID-19, is accompanied by mild symptoms with fever, cough, myalgia, fatigue, mild dyspnoea, sore throat, and headache. Nevertheless, great variability of symptoms from one person to another has been reported. Since COVID-19 is an emerging infectious disease, an additional problem is that the long-term effects and sequelae of the disease are unknown, both for moderate and severe forms [6]. Moreover, the psychological component must be considered, because it can condition many biological responses. In this regard, i.e., in the context of disasters and traumatic events, significant increases in stress levels have been observed in the population [7,8]. The same results were obtained from the first countries affected by the SARS-CoV-2 virus, such as China and South Korea [9,10]. The state of stress resulting from the pandemic and restrictive measures has long-term health effects, with an increased risk of physical and mental illness affecting sportsmen and sportswomen as well [11,12,13,14,15]. Even in the absence of pandemic situations, chronic stress is a major public health concern [16]. Moreover, it should be kept in mind that SARS-CoV-2 infection not only affects the pulmonary and heart systems, but also other organs such as the liver and kidneys [17,18,19]. All these disturbances must have an impact on sports performance, particularly among professional sportsmen and sportswomen [20]. In this line, physical performance is a complex concept that takes into account many aspects, including [21]: (a) efficient energy production (aerobic and anaerobic); (b) neuromuscular dynamics (strength and technical skills); and (c) psychological features (motivation and strategies). The COVID-19 pandemic has disturbed this framework.

In this narrative review, we have consulted publications from medical databases (mainly PubMed and Medline) reporting the systemic repercussions of COVID-19. Keywords used are mainly: COVID-19/SARS-CoV-2 and Sport. The data provided refer mainly to short-term problems undergone by athletes who have suffered the pathology. It is important to point out that COVID-19 is a very recent pathology and no conclusive long-term data are available. We have focussed on key aspects related to sports performance [22,23,24,25,26] and proposed general guidelines to make a healthy return mainly to professional physical activity from which many reports have been published [1,25,27,28,29,30]. In addition, we have to point out that for us a professional is an athlete that receives a month’s salary for training and competing. An amateur is mainly a person that does physical activity and competition out of their working time. Nevertheless, in the actual social context, the borders of such definitions are not entirely clear. We have focussed on professional athletes because they are under more strict control and data are available through many scientific publications.

## 2. Respiratory Disturbances

It has been well established that reduction of physical activity impairs correct glycemic control and changes body composition, favoring the increase in fat mass (FM) and reducing muscle mass (MM), all with negative consequences on maximal oxygen uptake (VO_2_max) [31,32]. This situation could be aggravated in athletes that have undergone SARS-CoV-2 infection [33]. In addition to the metabolic changes and psychological impact due to inactivity, respiratory alterations have to be considered. Many athletes reported residual symptoms even months after the initial COVID infection, including a persistent cough, tachycardia, and fatigue. This situation makes it difficult to return to physical activity due to the sustained demand of the respiratory system for optimal sports performance, particularly in aerobic disciplines [23]. In addition, maintaining a relatively stable but at the same time adapted metabolism during exercise poses a major challenge to respiratory and circulatory functions [34]. Therefore, programmed physical activity is instrumental during post-infection recovery in order to improve oxygen uptake, healthy circulating metabolic parameters, energy balance, and metabolic control. However, safe recommendations under physician supervision and in base on the new evidence have to be updated for athletes who have suffered the disease when returning to competition. Altogether, COVID-19 has an impact on metabolic adaptation during exercise [35]. Therefore, athletes with COVID-19 disease display an increased risk of reduced maximal and submaximal performance as well as altered cardiovascular and muscle metabolic adaptations [36] (Figure 1).

## 3. Muscular Repercussions

SARS-CoV-2 is capable of infecting multiple cell types with a preference for pulmonary epithelium and immune cells [37]. As a result, muscle pain (myalgia) and fatigue are usual initial symptoms of the disease, occurring in 35% of COVID-19 patients [38,39]. In general, these findings have been associated with critically ill myopathy and steroid myopathy associated with the clinical picture of COVID-19. This muscle deterioration could be explained through the activation of angiotensin-converting enzyme 2 (ACE2), a membrane-attached protein that mediates the entry of SARS-CoV-2. In this context, studies performed in animal models show that activation of ACE2 induces skeletal muscle alterations and reduces exercise capacity, with mitochondrial dysfunction and decreased oxidative fiber number, resulting in subsequent muscle atrophy [40].

Several factors may play a key role in muscle plasticity. One of them seems to be the degree of mechanical loading [41]. In this context, inactivity decreases the mass and size of muscle fibers and consequently leads to weakening [42]. Thus, it can be hypothesized that the acute inflammatory response to COVID-19 infection would consume the proteins that work as building blocks for muscle activity. As documented in other inflammatory processes, the synthesis of acute-phase proteins, such as C-reactive protein (CRP), ferritin, tumor necrosis factor-α (TNF-α), and the different interleukins (ILs), could appear concomitantly with albumin and muscle protein degradation. Nevertheless, this hypothesis needs to be investigated in future research.

Myalgia and fatigue are frequently observed in 44–70% of patients with COVID-19 [43], being the fifth most common symptom in patients with COVID-19 [38]. Several studies in patients infected with other coronaviruses have shown that myalgia increased serum CK concentrations [44] or produce rhabdomyolysis [45] in 1/3 of infected individuals. However, in the case of COVID-19, the circulating CK levels were close to baseline values (around 200 U/L), making it impossible to differentiate between a myogenic or neurogenic alteration [46,47]. It can be hypothesized that the “cytokine storm” induced by SARS-CoV-2 could be a possible mechanism underlying the persistent myalgia and fatigue [48], but this statement needs further research.

Under normal physiological conditions, intense and sustained exercise causes muscle damage with the release of muscle proteins leading to inflammation of myocytes and altered muscle integrity. This inflammatory response leads to increases in circulating muscle proteins, such as CK, lactate dehydrogenase (LDH), and myoglobin (Mb) as well as in pro-inflammatory cytokines, including TNF-α and IL-6 [49,50,51]. In the context of COVID-19, a similar inflammatory reaction occurs, together with high levels of acute phase reactants such as ferritin and CRP [52]. Increased pro-inflammatory cytokines are involved in the induction and effector phases of all immune and inflammatory responses [53]. In addition, the stress component that occurs in this situation enhances the synthesis of glucocorticoids. In this context, it exists a relationship between the immune system function, the inflammatory response, and the hypothalamic-pituitary-adrenal (HPA) system [54,55,56].

The muscle inflammatory process is associated with increased wasting, loss of strength, and functional damage [57]. One of the consequences is increased muscle fatigability. Fatigue is accompanied by the release of proteins and enzymes into the bloodstream: CK, Mb, and LDH. These proteins are indicators of muscle damage and muscle stress associated with intense exercise [58] (Figure 2). In the field of sport, it is known that exercise increases the activity of enzymes involved in glycolysis and glycogenolysis such as glycogen phosphorylase, phosphofructokinase, and LDH [59,60]. Therefore, an interesting hypothesis to check would be to investigate if the cumulative tiredness and metabolic demand that occurs during COVID-19 could be similar to the fatigue situation undergone after very intense exercise execution. Izquierdo et al. [61] observed that after a period of short-term strength training, exercise-induced loss of functional capacity occurred in athletes.

The review of Bogdanis [62] reported that muscle can react in different ways to be adapted to exercise demands, such as increasing size, variations in fiber composition, increased enzyme activity, and altered muscle activation. If appropriate adaptations occur, muscle fatigue is reduced during exercise. However, during the pandemic lockdown, athletes were forced into inactivity or other training systems with different impacts on muscle performance (Figure 2). Some disciplines managed to keep some fitness and performance as in the case of Olympic swimmers [63]. However, different outcomes have been reported in other disciplines during COVID-19 lockdowns. For instance, elite handball players maintained lower limb explosive strength; meanwhile, aerobic performance was not preserved [5]. In the same context, elite soccer players preserved aerobic performance during the lockdown training but the competitive power level was negatively affected [64]. In addition, detraining leads to a decrease in the oxidative capacity of muscles, as well as in VO_2_max and oxidative enzyme activity [59]. This point needs to be addressed in future research.

## 4. Cardiac Consequences

It has been established that COVID-19 leads to cardiac and vascular complications. A possible link to the above-mentioned “cytokine storm” is hypothesized [65]. “Cytokine storm” occurs in the severe phase of COVID-19 and could lead to impaired cardiac function, presenting characteristics similar to those reported in classic forms of stress or catecholamine-induced cardiomyopathy [66]. As indicated, this is more noticeable in severe COVID-19 cases, particularly in subjects with comorbidities, such as hypertension, type 2 diabetes, and cardiovascular disorders [67]. In addition, it seems that SARS-CoV-2 may directly infect cardiomyocytes, causing myocarditis with acute and severe deterioration of cardiac function [67].

From the data reported so far by different media, many professional team sports players have been infected with SARS-CoV-2. According to some reports, athletes appear to be at a higher risk of developing myocarditis than the general population, although there is no evidence to support these claims. It is true that intense and sustained exercise may influence susceptibility to infection, depending on the intensity and duration of physical activity [49,50,68,69].

Myocarditis associated with COVID-19 has been reported in almost 1/5 of patients, with a 50% of survival rate [70]. Myocarditis has traditionally been considered the main cause of life-threatening ventricular arrhythmias in sportsmen [71,72,73]. Therefore, this incidence would support the hypothesis of the development of cardiomyopathies in athletes suffering from COVID-19. It is well established that after a long period of inactivity, aerobic capacity (according to VO_2_max) can decrease significantly, accompanied by an increase in heart rate [35,74]. These physiological changes lead to a decrease in muscle capillaries and a loss of sensitivity in the mechanisms that control body temperature [75,76]. As mentioned before, different outcomes have been reported by diverse aerobic training protocols during COVID-19 lockdown [5,63,64].

Taking into account the novelty and limited knowledge of COVID-19, the cardiomyopathy prevalence and clinical implications (acute and late) are largely unknown. Therefore, the incidence of myocardial affectation, which in many cases may be silent for a long period of time after the resolution of typical COVID-19 symptomatology, is also unknown.

## 5. Neurological Involvement

Neurological symptoms have generally been reported in 36.4% of COVID-19 cases, consisting of altered consciousness, headaches, ischaemic strokes, epileptic seizures, and an altered sense of smell and taste [46,77]. However, despite the banal nature of these symptoms, autoimmune diseases are not uncommon. Ocular muscle paralysis, Miller-Fisher syndrome, and Guillain-Barré syndrome have been reported [78,79,80,81]. Recently, many cases of Guillain-Barré syndrome have been observed in patients with COVID-19, and an immune-mediated post-infectious inflammatory process has been implicated as the main cause. Neurological disorders of COVID-19 patients occur together with myalgia/fatigue, indicating as mentioned before, muscle injury [82].

Direct affectation of the central nervous system (CNS) in COVID-19 cases has also been described [83]. Cerebral ischaemia, cerebrospinal fluid (CSF) inflammatory syndrome without pathogen detection, associated with neurological symptoms, such as acute meningoencephalitis, has been described [84]. However, a conclusive diagnosis of neurotrophy or a causal relationship is currently not meaningful. In COVID-19, both CNS and the peripheral nervous system (PNS) are affected. Common neurological symptomatology includes myalgia, headache, and general malaise. Virus invasion seems to be responsible for severe neuronal complications, immune reactions, or hypoxic metabolic changes [85,86].

The pathophysiological mechanisms could be related to the possibility of virus entrance into neurons. As indicated before for muscle cells, neuronal tissue seems to express ACE2, evidenced in neurons and glial cells of animal models [87,88]. Nevertheless, the expression of ACE2 appears to be very limited in the healthy brain. Data from other studies with coronaviruses have reported that the viruses spread in neuronal cells along synapses, and sometimes also in the brain stem [89,90]. Another suspected route of invasion is following intranasal inoculation via the olfactory nerve into the CNS [89,91]. In mouse models of infection, transnasal inoculation of SARS-CoV has demonstrated virus spread into the brain, brainstem, and spinal cord [90]. Virus spread between neurons by axonal transport has been demonstrated in an experimental model with the human coronavirus OC43 [92].

Like other viruses, coronaviruses gain access to the CNS via two routes: hematogenous and transneuronal [91,93,94]. As indicated before, the neuro-invasive capacity of SARS-CoV-2 is unknown, the hematogenous route could be facilitated by disruption of the blood–brain barrier (BBB) associated with the “cytokine storm”, slowing microcirculation at the capillary level or facilitating the infection of myeloid cells and subsequent dissemination through the CNS [91]. High fever and cytopenia could reflect the severe hyperinflammatory syndrome at the CNS in COVID-19 patients. In addition, elevated IL-6 and ferritin levels are predictors of a fatal evolution of the disease [95]. The massive release of cytokines, chemokines, and other inflammatory signals (“cytokine storm”) results in the breakdown of the BBB. The process is amplified by the subsequent activation of toll-like receptors on microglia and astrocytes, promoting neuro-inflammation and altering homeostasis [96,97]. One hypothesis supports that SARS-CoV-2 infection could trigger reactive astrogliosis and activate microglia. Neurotropic viruses, such as some coronaviruses, are able to activate macrophages, microglia, and astrocytes and induce a pro-inflammatory state in the CNS [98]. In fact, primary cultures of glial cells infected with coronaviruses secrete factors such as IL-6, IL-12, IL-15, and TNFα leading to a state of chronic inflammation, responsible for brain damage [99].

## 6. State of Stress

There is no doubt that the pandemic situation caused by COVID-19 has led to increased stress levels. Prolonged exposure to crisis-related stress is likely linked to long-term health effects, including an elevated risk for physical and mental illnesses [100]. Stress increases the brain levels of the neurotransmitters dopamine and noradrenaline. These neurotransmitters initiate the activation of two major signal transmission pathways: the immediate sympathetic nervous pathway and the slower endocrine HPA axis. The sympathetic nervous pathway stimulates immediately the release of catecholamines from the adrenal medulla and adrenergic nerve terminals. On the other hand, the slower endocrine HPA pathway (activated around 20–30 min after the onset of stress) sets a series of chain reactions, stimulating the production of corticotropin-releasing hormone (CRH) by the hypothalamus, which in turn stimulates the release of adrenocorticotropic hormone (ACTH) and ß-endorphin by the adenohypophysis. ACTH induces cortisol release from the adrenal cortex [59,101].

Thus, the stress response is regulated by the HPA axis, which responds to survive in inappropriate or dangerous situations. Excessive or inadequate function of the autonomic nervous system (ANS) and the adrenocortical system is detrimental to health. Prolonged stimulation of these systems may be due to constant intense physical exercise, anxiety, exposure to adverse environments, interpersonal conflicts, lifestyle changes, and related behaviors resulting from being under chronic stress [54,101]. The COVID-19 pandemic has changed the main part of psychosocial issues, compromising mental health. The prevention and control measures during lockdown gave rise to challenging factors that aggravated further problems and resulted in public health complications. Therefore the pandemic had physical, nutritional, and psychological consequences that affected the health status of athletes. Uncertainty regarding financial and sporting future impacted preparation and performance once athletes returned to competition [102,103].

In the stress response, there are two key aspects. On one hand, there is a release of extracellular messengers such as catecholamines, which increase heart rate and blood pressure. These mediators promote adaptation to acute stress and would be related to acute fatigue that occurs for example during a training session or competition, leading to a decrease in performance [104]. On the other hand, chronic elevation of the same messengers can cause physiopathological changes. For example in the cardiovascular system, they can lead in the long term to the possible development of atherosclerosis and increased risk of myocardial infarction [105]. In the case of COVID-19, functional health complications are aggravated by mental problems, such as depression and anxiety, challenging post-pandemic therapeutic strategies [106].

Many other syndromes can be triggered as a consequence of stress. The common experience of being “stressed” is based on the elevation of some of the key systems that give rise to allostatic overload: cortisol, sympathetic activity, pro-inflammatory cytokines, with a decrease in parasympathetic activity [104,107]. In addition, we must bear in mind that the immune response is mediated by endocrine neural circuits (stress hormones) and by paracrine-endocrine circuits specific to the immune system including cytokines. These molecules act as emergency signals of the immune system, integrating and coordinating local and systemic signaling during immune and inflammatory reactions [104]. Although we have just mentioned aspects of stress that are driving pathological situations, stress is inherent to life and is a mechanism that allows living organisms to adapt. In this context, certain measures reduced mental health risks in sportsmen/women during the pandemic lockdown. Adapted and regular exercise routines were considered key strategies to face the psychological negative effects caused by the forced lockdown during the COVID-19 pandemic [108,109]. However, maintaining an optimal physical status to sustain performance was not the only challenge to keep in mind for athletes during the COVID-19 lockdowns. Post-exercise recovery and optimal resting were also key factors in the COVID-19 routine of athletes [110].

## 7. Proposals for Reintegration into Sport Practice

There is no doubt that COVID-19 represents a special situation that has changed dramatically the way by which sportsmen, sportswomen, and sports entities communicate with their fans. The relationship of sportsmen and sportswomen with fans is a key aspect of sports motivation and thereby optimal performance. However, one of the main concerns of athletes that suffered COVID-19 is the timing to return to full physical exertion and contact again with fans [63]. Therefore, we propose a likely strategy for returning to training and competition after COVID-19 infection.

The resumption of physical and sporting activity should be staggered and progressive, both in terms of time and intensity of exercise. An increase of 50% in intensity at the beginning and a gradual increase of 10–30% in the subsequent weeks are recommended [111]. This should allow for a progressive adaptation of organs and systems. A rapid return can cause problems at the neuromuscular level due to insufficient control during dynamic movements being a major risk factor for injury [112], particularly in flexibility-related actions [113]. Therefore, elite athletes can reach high-intensity training while avoiding sudden increases in training load. Nevertheless, a definitive algorithm has not been established yet and more research is necessary to update the proposed framework.

Athletes that have suffered COVID-19 can return to activity once medical diagnosis indicates no pulmonary or cardiac symptoms. One to two weeks of resting is recommended, depending on the cases that were asymptomatic or mild in sportsmen/women. A negative antigen test is mandatory before returning to activity. Several tests under the supervision of a medical doctor are recommended, including [11,25,114,115]: blood test (controlling levels of cardiac markers such as troponin and creatine kinase); electrocardiography, and echocardiography (to discard myocarditis and myopericarditis); 24–48 h Holter monitoring (to discard arrhythmias); lung functional test (to check pulmonary pathology); maximal exercise test (to establish physical fitness). Finally, psychological help could be necessary for particular circumstances or sports disciplines [1].

Obviously, it is compulsory to follow the rules and guidelines indicated by the health authorities in terms of hygiene, the distance among others, as well as those concerning sports facilities. Otherwise said, sports practice must be safe and healthy, in order to reduce the risk of infection.

## Figures and Tables

**Figure 1 ijerph-19-05400-f001:**
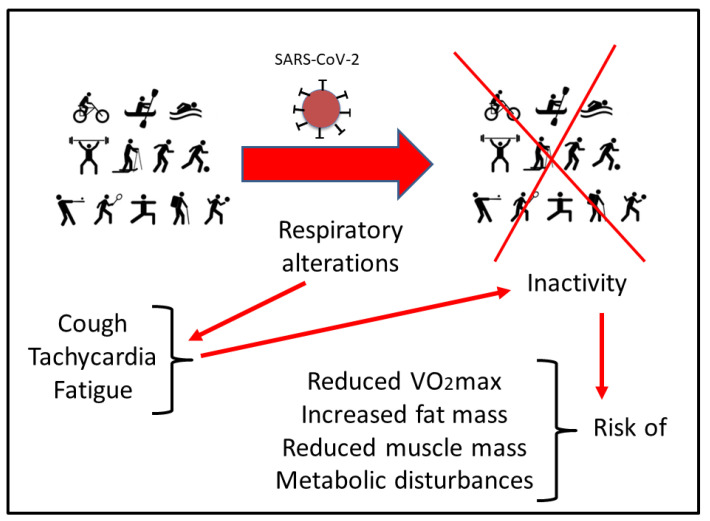
Scheme of the possible risks that can derive from the residual symptoms in respiratory function after SARS-CoV-2 infection. See the text for more details.

**Figure 2 ijerph-19-05400-f002:**
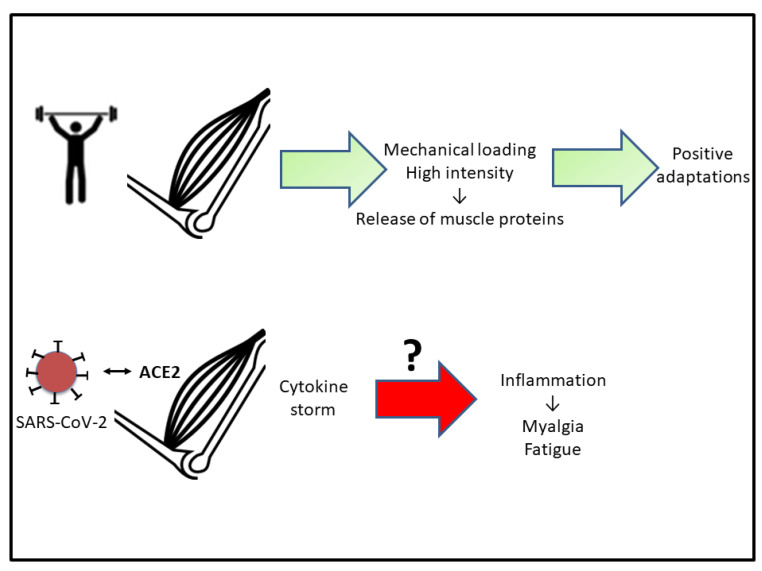
Scheme indicating that adequate physical activity results in positive muscle adaptations. The hypothesis to test (?) is if the inflammation associated with COVID-19 is responsible for myalgia and muscle fatigue. See the text for more details. Abbreviations used: ACE2, angiotensin-converting enzyme 2.

## Data Availability

Not applicable.

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
