# Peer review of "Effects and Causes of Detraining in Athletes Due to COVID-19: A Review"

_ijerph, 2022, doi:10.3390/ijerph19095400_

Round 1

Reviewer 1 Report

Dear Authors,

the title of your manuscript is “Clinical Impact of COVID-19 in Athletes Who Had Suffered 2 the Disease” and you describe it as a narrative review reporting systemic repercussions of COVID-19. The data provided refer mainly to short-term problems underwent by athletes who have suffered the pathology. Focussed on key aspects related to sports performance and proposing general guidelines to make a healthy return to professional physical activity.

However, only 35 references out of 102 dealt with COVID-19 and only 2 of them with infected athletes. Therefore, main changes are needed to address the title and aim proposed. Below you can find point by point comments.

Lines 2-3: since only 35 references out of 102 deal with COVID-19 and only 2 of them deal with athletes, if more appropriate references and discussions cannot be reported the title should be changed in: Effects and causes of detraining: a review.

Line 38: delete “to”

the cited reference (Guthold 2018) focus on insufficient physical activity, more specific effects of restrictions due to COVID-19 on athletes’ performance have been reported, here are some examples:

https://doi.org/10.3390/ijerph17228648

https://doi.org/10.3390/ijerph19042110

https://doi.org/10.23736/S0022-4707.20.11669-4

https://doi.org/10.1519/JSC.0000000000004028

https://doi.org/10.5114/biolsport.2021.109952

Lines 44-61: benefit of exercise on health is not the focus of this work, these paragraphs should be eliminated.

Lines 68-71: only reference 9 is specific to COVID 19 effect, here an example of more appropriate references on lockdown effects on athletes:

https://doi.org/10.3390/ijerph182312780

https://doi.org/10.1080/00913847.2020.1807297

https://doi.org/10.3389/fspor.2021.603415

https://doi.org/10.23736/S0022-4707.21.12401-6

https://doi.org/10.1080/24733938.2021.1933156

Lines 72-74: examples of references more related to COVID-19 consequences on health:

https://doi.org/10.1007/s11357-020-00272-3

https://doi.org/10.1007/s40520-020-01601-4

https://doi.org/10.1016/j.jad.2021.11.031

https://doi.org/10.4103/atm.atm_82_21

Line 76: change “keep” with “kept”

Line 88: change “underwent” with “undergone”

Lines 87-89: none of the reported reference deal with infected athletes, some examples of works on the topic:

https://doi.org/10.1080/24733938.2021.1933156

https://doi.org/10.1016/j.ijcard.2020.11.039

https://doi.org/10.1177/1941738120918876

https://doi.org/10.1001/jamacardio.2020.2136

https://doi.org/10.1136/bjsports-2020-102710

https://doi.org/10.1136/bjsports-2020-102482

https://doi.org/10.1001/jamacardio.2021.0565

Lines 90-92: guidelines for sport resumption have already reported, here are some examples:

https://doi.org/10.3390/ijerph17228648

https://doi.org/10.1136/bjsports-2020-102482

https://doi.org/10.1136/bmj.m4721

https://doi.org/10.1093/eurheartj/ehaa448

https://doi.org/10.1001/jamacardio.2020.5345

lines 98-125: references are related to the consequences of general detraining, not specifically due to COVID -19 restrictions, therefore, either the term COVID-19 is delated from the title or more specific references are needed.

Lines 135-138 and Figure 1: reported references are on previous respiratory infections, that COVID-19 could have the same effects can only be speculated, the statement and figure label must be corrected.

Lines 138-143: references 34 and 35 report results of previous respiratory infections or general detraining effects, this must be clarified when they are discussed.

Lines 148-149 and Figure 2: reference (Gu et al. 2005) reported results on previous respiratory infections, a direct connection between SARS-CoV-2 can only be speculated, the statement, figure and label must be corrected consequently.

Lines 164-170: none of the references dealt with COVID-19, a direct connection between SARS-CoV-2 and muscle damage can only be speculated, the discussion must address that it can be only hypothesised.

Lines 172-177: references 43, 45 and 46 do not deal with COVID-19, a direct connection between SARS-CoV-2 and increase in CK can only be speculated, it must be specified in the discussion. Only references 37 and 44 are related to the consequences of COVID -19 restrictions, therefore, either the term COVID-19 is delated from the title or more specific references are needed.

Lines 197-207: the statement “the cumulative fatigue and metabolic demand that occurs during COVID-19 are similar to the situation undergone after very intense exercise execution” is not supported by direct measurement or results reported in literature, it can only be presented as a hypothesis.

Lines 209-226 and 250-256: here a total absence of training is speculated, while some categories of athletes managed to keep some fitness and performance ability even during lockdown. Different outcomes have been reported depending on the level of athletes, individual or team sports, force or endurance training, this needs to be addressed in the discussion, see for example:

https://doi.org/10.3390/ijerph19042110

https://doi.org/10.5114/biolsport.2021.109952

https://doi.org/10.1055/a-1345-9262

Lines 239-240: the statement “From the data reported so far by different sports federations, many professional team sports players have been infected with SARS-CoV-2” is not supported by evidence, appropriate references are needed.

Lines 312-353: only reference 102 deals with stress as consequence of COVID-19, reference reporting the connection are needed, here are some examples:

https://doi.org/10.1016/j.heliyon.2020.e04315

https://doi.org/10.3390/ijerph17249419

https://doi.org/10.1016/j.jsams.2020.05.016

https://doi.org/10.1371/journal.pone.024020

https://doi.org/10.1177/0020764020925835

https://doi.org/10.3389/fpsyg.2021.669119

https://doi.org/10.3390/ijerph17186471

Lines 353-357: you need to support your statement; this aspect has been addressed in Olympic swimmers:

https://doi.org/doi.org/10.3390/ijerph19042110

Lines 360-369: those guidelines are generic precautions undertaken by all athletes resuming training after a period of interruption, references 107 and 108 do not refer to training restrictions due to COVID-19. As stated in your Reference 7 “Given the extreme mutability and unpredictability of the situation and the variety of clinical conditions caused by COVID19, we cannot know which subjects could be burdened by these consequences” and Reference 17 “Our knowledge of the effects of COVID-19 infection on physiological responses to exercise is very scarce. Research in this field will have to be carried to document this issue and update the proposed framework”. Guidelines for return to sports after COVID-19 are already available, see for example:

https://doi.org/10.3390/ijerph17228648

https://doi.org/10.1001/jamacardio.2020.5348

https://doi.org/10.24875/ACM.20000507

https://doi.org/10.1136/bjsports-2020-102482

https://doi.org/10.1714/3386.33637

Author Response

ANSWER TO REVIEWER-1

The title of your manuscript is “Clinical Impact of COVID-19 in Athletes Who Had Suffered the Disease” and you describe it as a narrative review reporting systemic repercussions of COVID-19. The data provided refer mainly to short-term problems underwent by athletes who have suffered the pathology. Focussed on key aspects related to sports performance and proposing general guidelines to make a healthy return to professional physical activity.

However, only 35 references out of 102 dealt with COVID-19 and only 2 of them with infected athletes. Therefore, main changes are needed to address the title and aim proposed. Below you can find point by point comments.

ANSWER: We thanks the Reviewer for the very good comments and the new references proposed that we have taken into account.

Lines 2-3: since only 35 references out of 102 deal with COVID-19 and only 2 of them deal with athletes, if more appropriate references and discussions cannot be reported the title should be changed in: Effects and causes of detraining: a review.

ANSWER: Taken into account the new references proposed, we have changed the title according to the suggestion made by the Reviewer (see lane 2).

Line 38: delete “to”.

ANSWER: “to” has been deleted (see lane 37).

The cited reference (Guthold 2018) focus on insufficient physical activity, more specific effects of restrictions due to COVID-19 on athletes’ performance have been reported, here are some examples:

https://doi.org/10.3390/ijerph17228648

https://doi.org/10.3390/ijerph19042110

https://doi.org/10.23736/S0022-4707.20.11669-4

https://doi.org/10.1519/JSC.0000000000004028

https://doi.org/10.5114/biolsport.2021.109952

ANSWER: The Reviewer is right. The provided references are more specific of the topic of this review. References are cited. See lane 41 and the new references 1-5.

Lines 44-61: benefit of exercise on health is not the focus of this work, these paragraphs should be eliminated.

ANSWER: Paragraphs have been deleted according to Reviewer suggestions.

Lines 68-71: only reference 9 is specific to COVID 19 effect, here an example of more appropriate references on lockdown effects on athletes:

https://doi.org/10.3390/ijerph182312780

https://doi.org/10.1080/00913847.2020.1807297

https://doi.org/10.3389/fspor.2021.603415

https://doi.org/10.23736/S0022-4707.21.12401-6

https://doi.org/10.1080/24733938.2021.1933156

ANSWER: Suggested references have been included. See lanes 54-55 and new references 11-15.

Lines 72-74: examples of references more related to COVID-19 consequences on health:

https://doi.org/10.1007/s11357-020-00272-3

https://doi.org/10.1007/s40520-020-01601-4

https://doi.org/10.1016/j.jad.2021.11.031

https://doi.org/10.4103/atm.atm_82_21

ANSWER: Suggested references have been included. See lane 58 and new references 17-19.

Line 76: change “keep” with “kept”

ANSWER: The change has been done (see lane 56).

Line 88: change “underwent” with “undergone”

ANSWER: The change has been done (see lane 68).

Lines 87-89: none of the reported reference deal with infected athletes, some examples of works on the topic:

https://doi.org/10.1080/24733938.2021.1933156

https://doi.org/10.1016/j.ijcard.2020.11.039

https://doi.org/10.1177/1941738120918876

https://doi.org/10.1001/jamacardio.2020.2136

https://doi.org/10.1136/bjsports-2020-102710

https://doi.org/10.1136/bjsports-2020-102482

https://doi.org/10.1001/jamacardio.2021.0565

ANSWER: Suggested references have been included. See lane 70-71 and new references 22-26.

Lines 90-92: guidelines for sport resumption have already reported, here are some examples:

https://doi.org/10.3390/ijerph17228648

https://doi.org/10.1136/bjsports-2020-102482

https://doi.org/10.1136/bmj.m4721

https://doi.org/10.1093/eurheartj/ehaa448

https://doi.org/10.1001/jamacardio.2020.5345

ANSWER: Suggested references have been included. See lane 72 and new references 27-30.

Lines 98-125: references are related to the consequences of general detraining, not specifically due to COVID -19 restrictions, therefore, either the term COVID-19 is deleted from the title or more specific references are needed.

ANSWER: A more specific description has been implemented regarding respiratory disturbances in athletes due to SARS-CoV-2 infection following Reviewer suggestions. See lanes 75-94.

Lines 135-138 and Figure 1: reported references are on previous respiratory infections, that COVID-19 could have the same effects can only be speculated, the statement and figure label must be corrected.

ANSWER: Figure 1 has been adapted accordingly.

Lines 138-143: references 34 and 35 report results of previous respiratory infections or general detraining effects, this must be clarified when they are discussed.

ANSWER: These contents have been eliminated. The text is only focussed in the respiratory disturbances in athletes as a result of COVID-19. See lanes 75-94 and new Figure 1.

Lines 148-149 and Figure 2: reference (Gu et al. 2005) reported results on previous respiratory infections, a direct connection between SARS-CoV-2 can only be speculated, the statement, figure and label must be corrected consequently.

ANSWER: References 37-39 serve as introductory in the corrected version of this section (see lanes 102-111). Figure 2 has been changed accordingly.

Lines 164-170: none of the references dealt with COVID-19, a direct connection between SARS-CoV-2 and muscle damage can only be speculated, the discussion must address that it can be only hypothesised.

ANSWER: The sentences have presented as a hypothesis that needs further investigation (see lanes 115, 117, 120-121).

Lines 172-177: references 43, 45 and 46 do not deal with COVID-19, a direct connection between SARS-CoV-2 and increase in CK can only be speculated, it must be specified in the discussion. Only references 37 and 44 are related to the consequences of COVID -19 restrictions, therefore, either the term COVID-19 is delated from the title or more specific references are needed.

ANSWER: The paragraph has been redefined, indicating that circulating CK values in COVID-19 are not so high as documented in other infections. Nevertheless, we open the possibility that myalgia could be related to the cytokine storm, but this needs further research (see lanes 123, 124, 127,129 and 131).

Lines 197-207: the statement “the cumulative fatigue and metabolic demand that occurs during COVID-19 are similar to the situation undergone after very intense exercise execution” is not supported by direct measurement or results reported in literature, it can only be presented as a hypothesis.

ANSWER: The statement has been presented as a hypothesis according to Reviewer suggestion (see lanes 153-155).

Lines 209-226 and 250-256: here a total absence of training is speculated, while some categories of athletes managed to keep some fitness and performance ability even during lockdown. Different outcomes have been reported depending on the level of athletes, individual or team sports, force or endurance training, this needs to be addressed in the discussion, see for example:

https://doi.org/10.3390/ijerph19042110

https://doi.org/10.5114/biolsport.2021.109952

https://doi.org/10.1055/a-1345-9262

ANSWER: The paragraph has been changed according to Reviewer suggestions (see lanes 167-179 and 207-209).

Lines 239-240: the statement “From the data reported so far by different sports federations, many professional team sports players have been infected with SARS-CoV-2” is not supported by evidence, appropriate references are needed.

ANSWER: This information was provided by the different media, coming in many cases from Sport Clubs and Federations. Therefore, the sentence has been changed accordingly.

Lines 312-353: only reference 102 deals with stress as consequence of COVID-19, reference reporting the connection are needed, here are some examples:

https://doi.org/10.1016/j.heliyon.2020.e04315

https://doi.org/10.3390/ijerph17249419

https://doi.org/10.1016/j.jsams.2020.05.016

https://doi.org/10.1371/journal.pone.024020

https://doi.org/10.1177/0020764020925835

https://doi.org/10.3389/fpsyg.2021.669119

https://doi.org/10.3390/ijerph17186471

ANSWER: This section of the review is focussed in molecular aspects related to stress. Nevertheless, we have completed this information with psychological aspects provided in the references indicated by the Reviewer that completed the presented evidences (see lanes 286-292, 301-303 and 314-321). See new references: 102, 103, 106, 108-110.

Lines 353-357: you need to support your statement; this aspect has been addressed in Olympic swimmers:

https://doi.org/doi.org/10.3390/ijerph19042110

ANSWER: This is supported by reference 63 (see lane 329).  

Lines 360-369: those guidelines are generic precautions undertaken by all athletes resuming training after a period of interruption, references 107 and 108 do not refer to training restrictions due to COVID-19. As stated in your Reference 7 “Given the extreme mutability and unpredictability of the situation and the variety of clinical conditions caused by COVID19, we cannot know which subjects could be burdened by these consequences” and Reference 17 “Our knowledge of the effects of COVID-19 infection on physiological responses to exercise is very scarce. Research in this field will have to be carried to document this issue and update the proposed framework”. Guidelines for return to sports after COVID-19 are already available, see for example:

https://doi.org/10.3390/ijerph17228648

https://doi.org/10.1001/jamacardio.2020.5348

https://doi.org/10.24875/ACM.20000507

https://doi.org/10.1136/bjsports-2020-102482

https://doi.org/10.1714/3386.33637

ANSWER: More specific guidelines have included (see lanes 332-352). New references have been cited: 114 and 115.

Reviewer 2 Report

In my opinion, this is an interesting topic. Covid-19 impacted the world population significantly. Therefore it is worth investigating its effects in various groups. In this term, especially interesting are athletes, who mainly state young and healthy people; however, there is a possible influence on health and physical performance.

In the abstract, the sentence: "We believe that the most important element  to take into account is the neuromuscular aspect, due to the implications that this system entails in  motion execution and coordination." It Sounds like your hypothesis. I think there should be a conclusion from Your review. Please consider rewriting.

I have a suggestion to remove or rewrite the paragraph from lines 81-84. In this form, it is not fit well with the rest of the introduction.

Moreover, I miss the information; how do you select the papers that consider covid-19 for Your manuscript? It would be best if you had to state your criteria. How did You define athlete – professional? Amateur? Any age consideration? Please provide more details.

I think the scheme of your writing is good. Firstly, you refer to the possible disturbances associated with covid-19; next, You provide the information, consider the general population, and next, when You want to refer to the athlete population. But this step, unfortunately, is a big issue in your paper. You do not provide enough adequate information. You refer to disease mechanism and theory and the general population, but there is a need to emphasize the information in the athletes' context. What has been found to date? I do not find the answer in Your review to what is known about the athletes who had suffered the disease (this words is in the title of Your study). The content of Your research is not fit the title. This is generally visible in all paragraphs. There is a lack of studies considering the athletes population.

Author Response

ANSWER TO REVIEWER-2

In my opinion, this is an interesting topic. Covid-19 impacted the world population significantly. Therefore it is worth investigating its effects in various groups. In this term, especially interesting are athletes, who mainly state young and healthy people; however, there is a possible influence on health and physical performance.

ANSWER: The Reviewer is right. Athletes are a segment of population that deserves a particular attention among others. This is the main objective of this narrative review.

In the abstract, the sentence: "We believe that the most important element to take into account is the neuromuscular aspect, due to the implications that this system entails in motion execution and coordination." It Sounds like your hypothesis. I think there should be a conclusion from Your review. Please consider rewriting.

ANSWER: We agree that the sentence can be interpreted as a hypothesis. We have placed at the beginning of the Abstract (see lanes 21-24).

I have a suggestion to remove or rewrite the paragraph from lines 81-84. In this form, it is not fit well with the rest of the introduction.

ANSWER: The paragraph has been rewritten and connects better with the INTRODUCTION and the rest of the review (see lanes 61-64).

Moreover, I miss the information; how do you select the papers that consider covid-19 for Your manuscript? It would be best if you had to state your criteria. How did You define athlete – professional? Amateur? Any age consideration? Please provide more details.

ANSWER: In a narrative review, selection of information is carried out in the scientific data base by introducing keywords such as: COVID-19 and Sport, SARS-CoV-2 and Sport (see lanes 66-68). On the other hand, for us a professional is an athlete that receives a month salary for training and compete. An amateur is mainly a person that does physical activity and competition out of its working time. Nevertheless, in the actual social context, the borders of such definitions are not entirely clear. We have focussed in professional athletes, because they are under a more strict control and data are available through publications (see lanes 72-73).

I think the scheme of your writing is good. Firstly, you refer to the possible disturbances associated with covid-19; next, you provide the information, consider the general population, and next, when You want to refer to the athlete population. But this step, unfortunately, is a big issue in your paper. You do not provide enough adequate information. You refer to disease mechanism and theory and the general population, but there is a need to emphasize the information in the athletes' context. What has been found to date? I do not find the answer in your review to what is known about the athletes who had suffered the disease (this words is in the title of Your study). The content of your research is not fit the title. This is generally visible in all paragraphs. There is a lack of studies considering the athletes population.

ANSWER: We have taken into account this important criticism made by the Reviewer. Now, all sections present information regarding the functional disturbances associated to COVID-19 in athletes. Section 2: lanes 75-94. Section 3: lanes 115-121, 150-157 and 168-177. Section 4: lanes 198-213. Section 5: We have found no consistent data regarding implications in athletes. For this reason, we have indicated some working hypothesis for future research. Section 6: lanes 286-292 and 314-321. Section 7: lanes 324-357. Specific references regarding COVID-19 and Sport are: 1-6, 11-15, 17-20, 22-30, 33, 36, 63, 64, 102, 103, 108-111, 114 and 115. TOTAL: 36 references.

Round 2

Reviewer 1 Report

Dear Authors,

I congratulate you on responding to all my comments, in the present form the work is well structured and presented.

Author Response

Thanks

Reviewer 2 Report

I appreciate the great work made by the authors. All issues were improved.

However, I have another suggestion:

The title was generally improved and fit better with the content of this study, but I think You should refer to the population on which you focus in this paper: for example:

“Effects and Causes of Detraining in Athletes Due to COVID-19: A Review”.

In my opinion, You should introduce in the text the athlete’s definition, which You adapted in Your paper. Just use the information provided to me in the answers.

I found  language issues. Below are some examples:

As mentioned before, different outcomes have been reported by different aerobic training protocols during COVID-19 lockdown” [5,63,64]. – you repeated “different” – use the synonym.

Line 22: “Spotswomen”; Line 33: doubled dot “..”; Line 78: “psichological”; Line 326: “recommeded”; 331: “circunstances

You need to check the whole text and improve all mistakes carefully!

Author Response

REVIEWER-2 (2nd round)

I appreciate the great work made by the authors. All issues were improved.

However, I have another suggestion:

The title was generally improved and fit better with the content of this study, but I think You should refer to the population on which you focus in this paper: for example:

“Effects and Causes of Detraining in Athletes Due to COVID-19: A Review”.

ANSWER: Title has been changed accordingly.

In my opinion, You should introduce in the text the athlete’s definition, which You adapted in Your paper. Just use the information provided to me in the answers.

ANSWER: Definition has been included as suggested. See lanes 74-80.

I found language issues. Below are some examples:

As mentioned before, different outcomes have been reported by different aerobic training protocols during COVID-19 lockdown” [5,63,64]. – you repeated “different” – use the synonym.

Line 22: “Spotswomen”; Line 33: doubled dot “..”; Line 78: “psichological”; Line 326: “recommeded”; 331: “circunstances

You need to check the whole text and improve all mistakes carefully!

ANSWER: The typos indicated by Reviewer have been corrected. See lanes 23, 87, 214, 354 and 359.  In any case, we go to ask to the Editorial team to check all language related issues.